# Curriculum Learning by Dynamic Instance Hardness

**Tianyi Zhou**[1]*, **Shengjie Wang**[1]*, **Jeff A. Bilmes**[2]
Paul G. Allen School of Computer Science & Engineering[1],
Department of Electrical & Computer Engineering[2], University of Washington, Seattle
{tianyizh, wangsj, bilmes}@uw.edu

## Abstract

A good teacher can adjust a curriculum based on students' learning history. By analogy, in this paper, we study the dynamics of a deep neural network's (DNN) performance on individual samples during its learning process. The observed properties allow us to develop an adaptive curriculum that leads to faster learning of more accurate models. We introduce *dynamic instance hardness* (**DIH**), the exponential moving average of a sample's instantaneous hardness (e.g., a loss, or a change in output) over the training history. A low DIH indicates that a model retains knowledge about a sample over time. For DNNs, we find that a sample's DIH early in training predicts its DIH in later stages. Hence, we can train a model using samples mostly with higher DIH and safely deprioritize those with lower DIH. This motivates a *DIH guided curriculum learning* (**DIHCL**) procedure. Compared to existing CL methods: (1) DIH is more stable over time than using only instantaneous hardness, which is noisy due to stochastic training and DNN's non-smoothness; (2) DIHCL is computationally inexpensive since it uses only a byproduct of back-propagation and thus does not require extra inference. On 11 datasets, DIHCL significantly outperforms random mini-batch SGD and recent CL methods in terms of efficiency and final performance. The code of DIHCL is available at https://github.com/tianyizhou/DIHCL.

## 1 Introduction

A curriculum plays an important role in human learning. Given different curricula of the same training materials, students' learning efficiency and performance can vary drastically. A good teacher is able to choose the contents of the next stage of learning according to a student's past performance. Analogously, in machine learning, instead of training the model with a random sequence of data, recent work in *curriculum learning* (CL) [4, 25, 17, 52, 13] shows that manipulating the sequence of training data can improve both training efficiency and model accuracy. In each epoch, CL selects a subset of training samples based on the difficulty and/or the informativeness of each sample — this is usually measured using *instantaneous feedback* from the model (e.g., the loss). CL then uses only these samples to update the model. Inspired by human learning curricula, a schedule of training samples is constructed (e.g., usually from easy to hard), sometimes combining with other criteria (e.g., diversity). As exhibited in previous work, CL can help to avoid local minima, improve the training efficiency, and can lead to better generalization performance.

Instantaneous hardness, however, does not take the training history of each sample into account. When applied to deep neural network (DNNs) training, and due to the non-smooth/non-convex nature of the loss and the randomness of stochastic gradient descent (SGD), the instantaneous hardness of each sample can change dramatically between consecutive epochs, so it is not reflective of the utility of each sample in the future. This results in a large difference between training sets selected over successive epochs, leading to an inconsistency of optimization objectives and gradients, and making

---

training less stable. Furthermore, keeping instantaneous hardness up to date requires extra inference steps of a model over all the samples, which can be expensive for DNNs [7, 15]. Though some recent work finds that data selection within each mini-batch [18] or based on the latest evaluated (but outdated) loss [28] may still perform well, this selection can be sub-optimal and unstable.

In this paper, we study the training dynamics of DNNs on individual samples from which a more accurate hardness measure can be computed that does not require extra inference and that can significantly improve performance. We study the difficulty a model has over time (i.e., training epochs) in learning each training sample. We introduce "*dynamic instance hardness* (**DIH**)" as the exponential moving average of an instantaneous hardness measure of a sample over time. We use three types of instantaneous hardness to compute DIH (fully defined in Section 2): the loss; the loss change; and the prediction flip (the 0-1 indicator of whether the prediction correctness changes) between two consecutive time steps. The first has been commonly used in CL, while the latter two capture a form of momentum of the loss/prediction.

We exploit several DIH properties that enable more effective CL approaches. Firstly, DIH can vary dramatically between different samples. Samples with smaller DIH seem to be more memorable (i.e., are retained more easily), while samples with larger DIH are harder to learn and retain. While the model is more likely to stay at a minimum of the easy samples' loss, its prediction on the hard samples is less stable under changes in optimization parameters (e.g., the learning rate). Secondly, unlike instantaneous hardness, the DIH status of a sample becomes consistent only after a few epochs. That is, a sample's DIH value converges quickly to its final relative position amongst all of the samples. For example, if a sample's DIH quickly becomes small, it stays small relative to the other samples; if it becomes large, it stays there. We can therefore accurately identify categories of hard and easy samples relatively early in the course of training. Thirdly, the DIH of each sample tends to monotonically decrease during training. This implies that the learning process strives for better local minimum for all samples, i.e., while easy samples stay easy throughout training, the hard samples also become easier the more we train on them.

These properties motivate a natural curriculum learning strategy "*DIH guided curriculum learning* (**DIHCL**)" that keeps training the model on those samples that have historically been hard since the model does not perform well on them. By contrast, it is safe to revisit easy samples (those with small DIH values) less frequently because the model is more likely to stay at those samples' minima. Hence, DIHCL helps a model focus on that which it finds difficult. This is similar to strategies that improve human learning, such as the Leitner system for spaced repetition [26]. This is also analogous to boosting [37] — in boosting, however, we average the instantaneous sample performance of multiple weak learners at the current time, while in DIHCL we average the instantaneous sample performance of one strong learner over the training history.

At each training step, DIHCL selects a subset of samples according to their DIH values, where the hard samples have higher probabilities of being selected relative to the easy samples. The model is updated by (stochastic) gradients computed on the selected samples. We then update the DIH of the selected samples by using their instantaneous hardness, a byproduct of back-propagation (since it needs to perform inference at first, e.g., a forward-propagation of a DNN). This significantly improves the efficiency of previous CL methods, which rely on extra inference steps to evaluate the instantaneous hardness of all the samples. Here, it is safe to update only the DIH of the selected samples since the unselected ones have smaller and decreasing DIH values (due to the observed properties of DIH) and thus keeping a stale DIH for them will not reduce their chance of being selected in the future steps. As mentioned earlier, in the training of DNNs, the hardness ranking of each sample by DIH will quickly converge after a few training steps and remains consistent for future steps. Those early steps provide the opportunity for the necessary exploration to ensure that hardness ranking is via DIH is accurate. To improve the exploration efficiency, DIHCL sweeps through the entire training set for the first few epochs and then starts to select training samples by DIH-weighted subset (random) sampling, and we gradually decrease the subset size during training. We provide several options for weighted sampling, using different distributions, and we integrate subset diversity into the selection criteria as well when feasible. Empirically, we evaluate several variants of DIHCL and compare them against random mini-batch SGD as well as recent curriculum learning algorithms on 11 datasets. DIHCL shows an advantage over other baselines in terms both of time/sample efficiency and test set accuracy.

## 1.1 Related Work

Early curriculum learning (CL) [20, 3, 40] work shows that feeding an optimized sequence of training sets (i.e., a curriculum), that can be designed by a human expert [4], into the training algorithms can improve the models' performance. Self-paced learning (SPL) [25, 42, 41, 43] chooses the curriculum based on hardness (e.g., per-sample loss) during training. SPL selects samples with smaller loss, and gradually increases the subset size over time to cover all the training data. Self-paced curriculum learning [17] combines the human expert in CL and loss-adaptation in SPL. SPL with diversity (SPLD) [16] adds a negative group sparse regularization term to SPL and increases its weight to increase selection diversity. Machine teaching [20, 55, 35] aims to find the optimal and smallest training subset in order to produce similar performance as when all the data is used. Minimax curriculum learning (MCL) [52] argues that the diversity of samples [47, 19, 46] is more critical in early learning since it encourages exploration, while completeness becomes more useful later. It also uses a form of instantaneous instance hardness (loss) but is not dynamic like DIH, and it formulates optimization as a minimax problem. Compared to the above methods, DIHCL has the following advantages: (1) DIHCL improves the efficiency of CL since extra inference on the entire training set per step is not required; and (2) DIHCL uses DIH as the metric for hardness which is a more stable measure than instantaneous hardness.

Previous works [39, 36, 38, 54] use "instance hardness" defined as $1 - p_w(y_i|x_i)$, i.e., the complement of the posterior probability of label $y_i$ given input $x_i$ for the $i^{\text{th}}$ sample under model $w$, which does not take the training dynamics into account. More recently, a special case of DIH has been studied in [45], which computes the mean of the prediction flips over all the steps after training has occurred. They show that removing samples with the smallest prediction flip average from the training set leads to less degradation of generalization performance than removing random samples. Based on this observation, they propose to train a small neural net beforehand to determine hard samples, which are then used to train a large neural net. By contrast, our study of DIH focuses on its dynamic properties **during** training, which inspires a novel curriculum learning strategy that can be applied to each step before training completes. A similar idea has been recently studied for semi-supervised learning [53]. Historical dynamics has been used to estimate prediction uncertainty over time in [7] and MentorNet [18]. However, the former treats all historical steps equally, while the latter still relies on the instantaneous difference of a loss to its historical moving average.

The training dynamics in this paper is also related to the memorization studied in [51], which considers overfitting on noisy data with random labels. We discuss this in the appendix (see Figure 9) showing that noisy data has distinctive training dynamics. Our observations also suggest that learning simple patterns [1] happens mainly from the easily memorable samples early during training. Our problem is distinct from catastrophic forgetting [21], which considers sequential learning of multiple tasks, where later learned tasks make the model

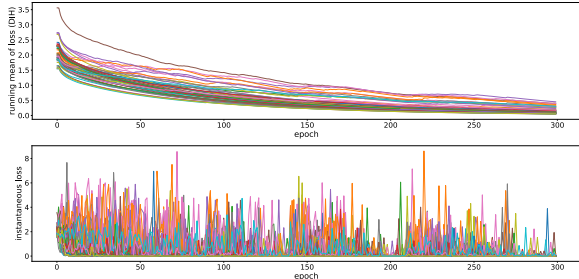

Figure 1: **Top:** DIH (running mean of loss) vs. **Bottom:** instantaneous loss of 50 randomly selected samples from CIFAR10 on WideResNet-28-10.

forget what has been learned from earlier tasks. In our work, we consider single task learning.

## 2 Dynamic Instance Hardness

Let $a_t(i)$ be a measure of instantaneous (i.e., at time $t$) hardness of a sample $(x_i, y_i)$ with feature $x_i$ and ground truth label $y_i$, where $i$ is a sample index and $t$ is training iteration (typically, a count of mini-batches that so far have been processed). We consider three different metrics of instantaneous instance hardness in this work:

(A) Loss evaluation $\ell(y_i, F(x_i; w_t))$, where $\ell(\cdot, \cdot)$ is a standard loss function and $F(\cdot; w)$ is the model where $w$ are the model parameters;

(B) Loss change $|\ell(y_i, F(x_i; w_t)) - \ell(y_i, F(x_i; w_{t-1}))|$ between two consecutive time steps;

(C) Prediction flip $|\mathbb{1}[\hat{y}_i^t = y_i] - \mathbb{1}[\hat{y}_i^{t-1} = y_i]|$, where $\hat{y}_i^t$ is the prediction of sample $i$ in step $t$, e.g., $\arg\max_j F(x_i; w_t)[j]$ for classification.

(A) corresponds closely to the "instance hardness" of [39]. However, (B) and (C) require information from previous time steps and aim to capture a form of momentum. Nevertheless, we consider (A),

(B), and (C) all to be variations of instantaneous instance hardness since they use information from only a local time window around iteration $t$. We define dynamic instance hardness (DIH) as a running average over an instantaneous instance hardness, defined and computed recursively as

$$r_{t+1}(i) = \begin{cases} \gamma \times a_t(i) + (1 - \gamma) \times r_t(i) & \text{if } i \in S_t \\ r_t(i) & \text{else ,} \end{cases} \quad (1)$$

where $\gamma \in [0, 1]$ is a discount factor, $S_t \subseteq V$, and $V = [n]$ is the set of all $n$ training sample indices. $S_t$ is the set of samples used for training at time $t$, e.g., a subset selected by some curriculum learning method (or a random batch in some cases). In general, $S_t$ should be large early in training, but as $r_t(i)$ decreases for many samples, choosing a smaller but wiser $S_t$ will result in faster training and more accurate models. The work of [45] uses a special case of DIH at $t = T$ ($T$ is the total number of training steps) in Eq. (1) with $\gamma = 1/t+1$, $S_t = V$, and $a_t(i)$ being prediction flips (case (C)).

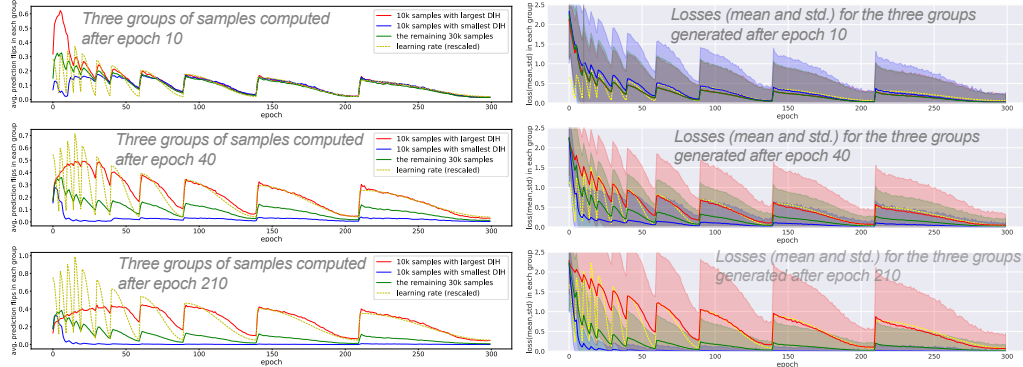

Figure 2: **LEFT:** Averaged prediction-flip and **RIGHT:** losses (mean and std.) of the three groups of samples partitioned by a DIH metric (i.e.,running mean of prediction-flip) computed at epoch 10 (top), 40 (middle), and 210 (bottom) during training a WideResNet-28-10 on CIFAR10. Early DIH (epoch 40) can already predict the forgettable and memorable samples at much later stages (epoch 210). The failed partition based on epoch 10 shows the importance of sufficient exploration needed for DIH to accurately measure hardness over time.

**Experimental setting:** When training DNNs, as shown in Figure 1, the instantaneous hardness measure (e.g., the per-sample loss) is usually too noisy and unstable to reflect the learning progress of the model. DIH, on the other hand, is a simple alternative descriptor of the training dynamics that averages out the noise. In the following, we use DIH as a tool to study the training dynamics of DNNs on individual samples. We train a WideResNet of depth 28 and width factor 10 on the CIFAR10 dataset by random mini-batch SGD, and apply a modified cosine annealing learning rate schedule [29] for multiple epochs of increasing length (300 epochs in total) and a decaying target learning rate. We contend that a cyclic learning rate suits our study because: (1) it includes the most commonly used monotone decreasing schedule since the learning rate in each cycle is decreasing; (2) compared to a monotone decreasing schedule, it can uncover the dynamic properties of DIH in more scenarios such as increasing learning rates and different learning rate decay speeds. In the study, we compute DIH using two types of instantaneous instance hardness, where $a_t(i)$ is either loss or prediction flips (i.e., cases (A) or (C)). Since we do not apply any curriculum learning just yet, we always keep $S_t = V = [50000]$.

Instead of visualizing $r_t(i)$ for all $i \in [50000]$ training samples, we use $r_t(i)$ (with $a_t(i)$ being prediction flips) to categorize them into three groups, and we do this at epochs 10 (early training), 40 (middle), and 210 (later training). At epoch 40, the 10,000 samples with the largest $r_{40}(i)$ comprise the first group, the 10,000 samples ones with the smallest $r_{40}(i)$ comprise the next group, and the remaining 30,000 samples comprise the final group. We will show that the training dynamics of the three groups have different characteristics. In Figure 2, we plot the dynamics of the average prediction flips over each group (left plot) and the mean/standard deviation of loss in each group (right plot).

**DNNs have very different training dynamics on samples with small and large DIH.** At any step-$t$, we observe (in our empirical studies) that a group of samples with small $r_t(i)$ are quickly learned in early epochs and, thereafter, their losses remain small with predictions almost unchanged. Since the behaviour on these samples is stationary even when the model changes by many steps with varying step sizes over the loss landscape along noisy SGD directions, this implies that the model reaches a point that is a relatively flat local minimum common amongst these samples. This suggests that it is safe to revisit these samples less frequently. By contrast, the samples with large $r_t(i)$ show a large variance during training, i.e., their losses oscillate between small and large values and their predictions frequently change, indicating difficulty. Their dynamics on average trace the changes of

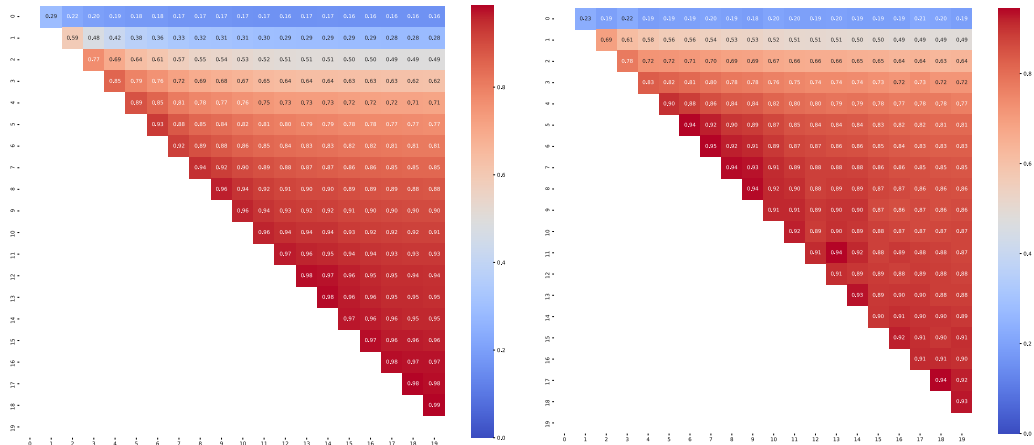

Figure 3: **LEFT:** Entry $A_{i,j}$ ($i < j$) is the percentage of shared samples between the 10k samples with the **largest** DIH computed in epoch $15i$ and epoch $15j$. **RIGHT:** Entry $A_{i,j}$ ($i < j$) is the percentage of shared samples between the 10k samples with the **smallest** DIH computed in epoch $15i$ and epoch $15j$. It shows that both the hard and easy samples in the future are predictable using the DIH values computed in early epochs.

learning rate, which implies that doing well on these samples is achieved only at relatively sharp local minima. This suggests that the model learns and generalizes better and faster on the easy samples than the hard ones. Similar to human learning [26], a natural strategy would learn the hard samples more frequently (to search for better local minima) while reducing the reencounter frequency of the easy, already learnt, ones. This can also reduce computation since it is focused more where it is needed.

We can use DIH to identify the easy and hard samples accurately, but do we need to pay the price of training a model until convergence like [45] in order to get the DIH values? By comparing the plots with different $t$ in Figure 2, we can see that **DIH in early epochs suffices to identify the easy vs. the hard samples**. The samples with small DIH at epoch 40 will remain relatively small compared to other samples even at later epochs. The hard (large DIH) samples remain hard in the future. That is, based on $r_t(i)$ (even for early stages when $t$ is not large), we surmise that it will be prudent to apply additional training effort on hard samples and begin de-emphasizing the already learnt easy samples.

We empirically verify that the samples with large/small DIH in the future can be predicted by only using the DIH during early epochs. In Figure 3, we show the overlap rate of hard/easy samples between pairs of epochs as two upper-triangle matrices. For example, given $U_i$, the 10k samples with the **largest** DIH in epoch $15i$, and $U_j$ for any $j > i$, $A_{i,j} = |U_i \cap U_j|/10000$ for the matrix $A$ in the left plot. Similarly, the matrix in the right plot measures the overlap rate for the 10k samples with the **smallest** DIH between epoch $15i$ and epoch $15j$. They show that after a few early epochs, DIH can accurately predict the hard and easy samples in the future. This verifies our statement in the last paragraph. In addition,

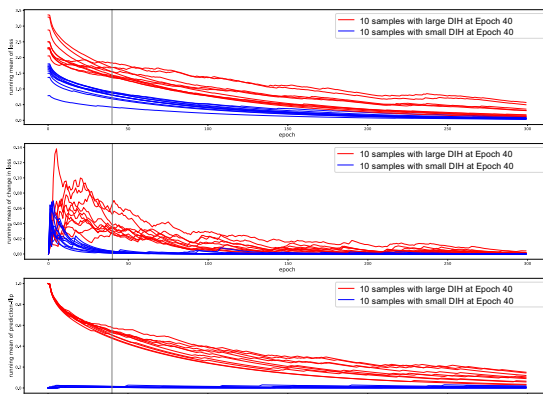

Figure 4: The three strategies for DIH on 10 hard and 10 easy samples, each that have been randomly sampled from the top 10k samples with the largest/smallest DIH at Epoch 40.

it shows that $|U_i \cap U_j|/10000$ between consecutive epochs $15i$ and $15j$ is close to $100\%$, which suggests that DIH is a stable, consistent, and smoothly varying measure. This allows us to save computation by lazily updating DIH only on a subset of samples $S_t$ per step during training, as we do in the definition of DIH in Eq. (1).

We find that **DIH on the same sample is robust and insensitive to the randomness of training.** To verify this, we run the above experiment twice on CIFAR10 using two different random seeds. We then compute the overlap rate between the 10k samples with the largest/smallest DIH at

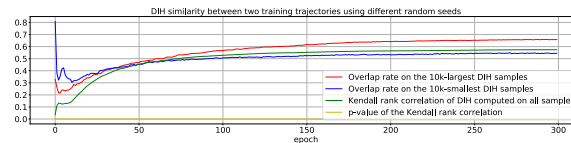

Figure 5: Correlation between DIH in two training experiments using different random seeds.

each epoch. Since DIH performs as a score to select/rank samples in our approach, we can use Kendall rank correlation coefficient (i.e., Kendall's $\tau$) to evaluate the correlation between DIH values of a sample at the same epoch of the two random experiments. We report the overlap rates and Kendall's $\tau$ (with its p-value) in Figure 5. During training, the two overlap rates and Kendall's $\tau$ all quickly grow to $> 0.6$ while the p-value stays near 0. This indicates a strong correlation between the DIH of the two random trials.

Previous CL methods using instantaneous hardness need to evaluate it for all samples before selecting any sample in each step, which involves extra inference computation for the unselected samples. This is expensive when training DNNs. While switching to DIH, such extra computation can be avoided. Figure 4 shows that all three types ((A), (B), and (C)) of **DIH metrics decrease during training for both easy and hard samples.** If our curriculum prefers selecting hard samples with large DIH, it is safe to update their DIH lazily, since the stale DIH values for the unselected samples are greater than their up-to-date true values, so the future steps will not miss informative samples. Updating DIH of the selected samples requires only a byproduct of back-propagation, which is already available after inference completes. This also indicates that as learning continues, samples become less informative, so that we can select and train on fewer samples.

In the appendix, we further report and compare the dynamics in four scenarios using plots as Figure 2: (1) under 100% label noise (Figure 9); (2) under 40% label noise (Figure 14); (3) training a smaller DNN (Figure 10); and (4) using exponential decaying learning rate across epochs (Figure 13). They show that (1) the dynamics revealed by DIH is entirely different for data with incorrect labels; (2) clean and noisy labeled data exhibit different dynamics that can be distinguished by DIH; (3) DIH values across samples are more different on deeper and wider DNNs, implying that large DNNs are more data-selective in learning; and (4) DIH shows similar properties with other learning rate schedules.

## 3 DIH guided Curriculum Learning

The above properties of DIH and training dynamics of DNNs naturally motivate a curriculum that trains DNNs using hard samples (those with large DIH) and reduces learning tendency on easy ones.

### 3.1 A "Free" Curriculum

We arrive at a curriculum learning strategy that selects harder samples to train the model at each step. And since only a subset of samples at each epoch are used to train on, and since DIH is updated only for the selected samples, we find that computation to achieve a given accuracy is reduced. We give a greedy version of DIHCL in Algorithm 1, where $\{(x_i, y_i)\}_{i=1}^n$ is the training data, $\pi(\cdot; \eta)$ is an optimization method such as SGD, $\eta_{1:T}$ are the $T$ learning rates, and $\gamma_k$ is the reduction factor for subset sizes $k_t$. DIHCL trains using more samples early on to produce an accurate initial estimate of $r_t(i)$. This is indicated by $T_0$, the number of warm start epochs over the whole training set. After iteration $T_0$, we gradually reduce the

---
**Algorithm 1** DIH Curriculum Learning (DIHCL-Greedy)
---
1: **input:** $\{(x_i, y_i)\}_{i=1}^n$, $\pi(\cdot; \eta)$, $\ell(\cdot, \cdot)$, $F(\cdot; w)$;
$\qquad$ $\eta_{1:T}$; $T, T_0$; $\gamma, \gamma_k \in [0, 1]$
2: **initialize:** $w, \eta_1$, $k_1 = n$, $r_0(i) = 1 \ \forall i \in [n]$
3: **for** $t \in \{1, \cdots, T\}$ **do**
4: $\quad$ **if** $t \le T_0$ **then**
5: $\quad\quad$ $S_t \leftarrow [n]$;
6: $\quad$ **else**
7: $\quad\quad$ Let $S_t = \text{argmax}_{S:|S|=k_t} \sum_{i \in S} r_t(i)$;
8: $\quad$ **end if**
9: $\quad$ Apply optimization $\pi(\cdot; \eta)$ to update model:
$$w_t \leftarrow w_{t-1} + \pi \left( \nabla_w \sum_{i \in S_t} \ell(y_i, F(x_i; w_{t-1})); \eta_t \right)$$
10: $\quad$ Compute normalized $a_t(i)$ for $i \in S_t$ using Eq. (2);
11: $\quad$ Update DIH $r_{t+1}(i)$ using Eq. (1);
12: $\quad$ $k_{t+1} \leftarrow \gamma_k \times k_t$;
13: **end for**
---

number of samples from $k_1 = n$ to $k_t$ thereby focusing on the most difficult samples as training proceeds. At step $t$, we select subset $S_t \subseteq [n]$ with large $r_{t-1}(i)$ and then update the model by training on $S_t$. We then update $r_t(i)$ via Eq. (1).

Since the learning rate can change over different steps, and large learning rates mean greater model change, we normalize $a_t(i)$ by the learning rate $\eta_{t-1}$[2]. Specifically, we apply one of the following

depending on which form of $a_t(i)$ we are using (case (A), (B), or (C) above):

$$(A) \ a_t(i) \leftarrow \ell(y_i, F(x_i; w_{t-1}))/\eta_t,$$
$$(B) \ a_t(i) \leftarrow |\ell(y_i, F(x_i; w_{t-1})) - \ell(y_i, F(x_i; w_{\tau_t(i)-1}))|/ \sum_{t'=\tau_t(i)}^{t} \eta_{t'}, \qquad (2)$$
$$(C) \ a_t(i) \leftarrow |\mathbb{1}[\hat{y}_i^t = y_i] - \mathbb{1}[\hat{y}_i^{t-1} = y_i]|/ \sum_{t'=\tau_t(i)}^{t} \eta_{t'},$$

where $\tau_t(i) < t - 1$ indicates the most recent step before $t-1$ when $i$ was selected. The $T_0$ warm start epochs and the schedule of decreasing $k_t$ are necessary for early exploration since DIH is a running mean over a sample's dynamics and thus needs to revisit each sample to estimate its relative DIH position. A simple method to further reduce training time in early stages is to extract and use only a small and diverse subset of $S_t$. Inspired by MCL [52], after line 7, we reduce $S_t$ to a subset of size $k_t' = \gamma_{k'} k_t$ ($0 < \gamma_{k'} \leq 1$) by (approximately) solving the following submodular maximization.

$$\max_{S \subseteq S_t, |S| \leq k_t'} \sum_{i \in S} r_t(i) + \lambda_t G(S) \qquad (3)$$

The function $G : 2^{S_t} \to \mathbb{R}_+$ can be any submodular function [12], and hence we can exploit fast greedy algorithms [33, 31, 32] to solve Eq. (3) with an approximation guarantee. We gradually reduce preference for diversity as training proceeds by reduce $\lambda_t$ by a factor $0 \leq \gamma_\lambda \leq 1$ at each step.

Table 1: The test accuracy (%) achieved by different methods training DNNs on 11 datasets (without pre-training). We use "Loss, dLoss, Flip" to denote the 3 choices of DIH metrics based on (A), (B), and (C) respectively. In all DIHCL variants, we apply diversity (and greedy submodular maximization using the lazier-than-lazy-greedy procedure [32]) for Eq. (3) on only the datasets CIFAR10, CIFAR100, STL10, SVHN, KMNIST, and FMNIST. In this case, the first $T_0$ warm-start epochs of DNN training was used to also produce the feature extractor $z(x)$ to instantiate the facility location function. The other datasets did not employ diversity, and we leave that to future work. For each dataset, the best accuracy is in blue, the second best is red, and third best is green.

| Curriculum | CIFAR10 | CIFAR100 | Food-101 | ImageNet | STL10 | SVHN | KMNIST | FMNIST | Birdsnap | Aircraft | Cars |
|---|---|---|---|---|---|---|---|---|---|---|---|
| Rand mini-batch | 96.18 | 79.64 | 83.56 | 75.04 | 86.06 | 96.48 | 98.67 | 95.22 | 64.23 | 74.71 | 78.73 |
| SPL | 93.55 | 80.25 | 81.36 | 73.23 | 81.33 | 96.15 | 97.24 | 92.09 | 63.26 | 68.95 | 77.61 |
| MCL | 96.60 | 80.99 | 84.18 | 75.09 | 88.57 | 96.93 | 99.09 | 95.07 | 65.76 | 75.28 | 76.98 |
| DIHCL-Rand, Loss | 96.76 | 80.77 | 83.82 | 75.41 | 87.25 | 96.81 | 99.10 | 95.69 | 65.62 | 79.00 | 80.91 |
| DIHCL-Rand, dLoss | 96.73 | 80.65 | 83.82 | 75.34 | 86.93 | 96.83 | 99.14 | 95.64 | 65.25 | 79.93 | 78.70 |
| DIHCL-Exp, Loss | 97.03 | 82.23 | 84.65 | 75.10 | 88.36 | 96.91 | 99.20 | 95.45 | 66.13 | 77.68 | 79.85 |
| DIHCL-Exp, dLoss | 96.40 | 81.42 | 84.75 | 75.62 | 89.41 | 96.80 | 99.18 | 95.50 | 66.59 | 79.72 | 81.48 |
| DIHCL-Beta, Flip | 96.51 | 81.06 | 84.94 | 76.33 | 86.88 | 97.18 | 99.05 | 95.66 | 65.48 | 78.49 | 80.13 |

## 3.2 Practical DIHCL using DIH-weighted Sampling

In line 7 of Alg. 1, we select $S_t$ with the highest $r_{t-1}(i)$ values. In practice, we find adding randomness to the selection procedure gives better performance as (1) exploration on samples with small $r_t(i)$ are necessary to accurately estimate to $r_t(i)$, and (2) randomness of training samples is essential to achieve a good quality solution $w$ for non-convex models such as DNNs. Instead of choosing the top $k_t$ samples greedily and deterministically, we perform a randomized greedy procedure by sampling with probability $p_{t,i} \propto h(r_{t-1}(i))$, where $h(\cdot)$ is a monotone non-decreasing function, similar to [27, 28]. Hence, we still prefer data points with high DIH. An ideal choice of $h(\cdot)$ should balance between the exploration (under poorly estimated DIH values) and exploitation (when DIH is well estimated). We propose the following three sampling methods to replace line 7 of Alg. 1, and give extensive evaluations in the experimental section.

**DIHCL-Rand:** Let $h(r_t(i)) = r_t(i)$. We sample data with probability proportional to DIH values.

**DIHCL-Exp:** We trade-off exploration and exploitation similarly to Exp3 [2], which samples based on the softmax value. We then reweigh the observation by the selection probability to encourage exploration: $h(r_t(i)) = \exp\left[\sqrt{2\log n/n} \times r_t(i)\right]$, $a_t(i) \leftarrow a_t(i)/p_{t,i}$ $\forall i \in S_t$.

**DIHCL-Beta:** We utilize the idea of Thompson sampling [44] and use a Beta prior distribution to balance exploration and exploitation, i.e., $h(r_t(i)) \sim \text{Beta}(r_t(i), c - r_t(i))$, where $c$ is a sufficiently large constant with $c \geq r_t(i)$, e.g., $c = 1$ when $a_t(i)$ is prediction flip. The Beta distribution encourages exploration when the difference between $r_t(i)$ and $c - r_t(i)$ is small.

## 4 Experiments

We train different DNNs by using variants of DIHCL, and compare them with three baselines, vanilla random mini-batch SGD, self-paced learning (SPL) [25], and minimax curriculum learning

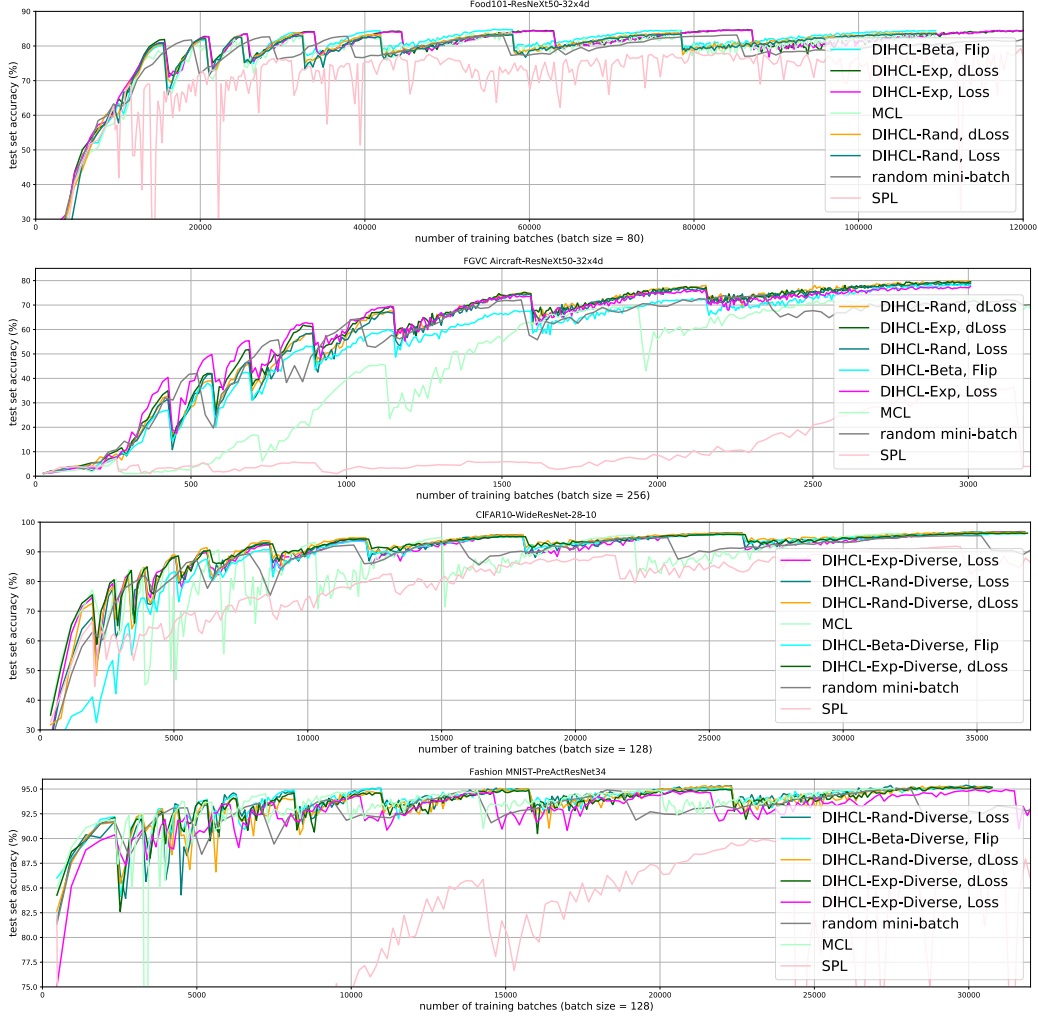

Figure 6: Training DNNs by using DIHCL (and its variants), SPL [25], MCL [52], and random mini-batch SGD on Food-101, Aircraft, CIFAR-10 and FMNIST. We use "Diverse" to denote DIHCL that further reduces $S_t$ by applying submodular maximization for Eq. (3). We report how test accuracy changes during training.

(MCL) [52] on 11 image classification datasets (without pre-training), i.e., **(A)** WideResNet-28-10 [50] on CIFAR10 and CIFAR100 [24]; **(B)** ResNeXt50-32x4d [49] on Food-101 [6], FGVC Aircraft (Aircraft) [30], Stanford Cars [23], and Birdsnap [5]; **(C)** ResNet50 [14] on ImageNet [11]; **(D)** WideResNet-16-8 on Fashion-MNIST (FMNIST) [48] and Kuzushiji-MNIST (KMNIST) [8]; **(E)** PreActResNet34 [14] on STL10 [9] and SVHN [34]. We use mini-batch SGD with a momentum of 0.9 and a cyclic cosine annealing learning rate schedule [29] (multiple epochs with starting/target learning rate decayed by a multiplicative factor 0.85). We use $T_0 = 5, \gamma = 0.95, \gamma_k = 0.85$ for all DIHCL variants, and gradually reduce $k$ from $n$ to $0.2n$. We chose $T_0 = 5$ since it is sufficient to produce a reasonably good model to estimate DIH. We tried $\gamma = 0.95, 0.9, 0, 8$ and they perform similarly, e.g., for DIHCL-Rand Loss, on CIFAR10, $\gamma = 0.95, 0.9, 0, 8$ lead to accuracy of $96.76\%, 96.75\%, 96.78\%$ respectively. We chose $\gamma_k = 0.85$ so we can reduce the size of $S_t$ from $n$ to $0.2n$ in 10 epochs. On each dataset, we apply each method to train the same model for the same number of epochs, but each method may select a different number of samples per epoch. More details about the datasets and settings can be found in the appendix. For DIHCL variants that further reduce $S_t$ by solving Eq. (3), we use $\lambda_1 = 1.0, \gamma_\lambda = 0.8, \gamma_{k'} = 0.4$ and employ the "facility location" submodular function [10] $G(S) = \sum_{j \in S_t} \max_{i \in S} \omega_{i,j}$ where $\omega_{i,j}$ represents the similarity between sample $x_i$ and $x_j$. We utilize a Gaussian kernel for similarity using neural net features (e.g., the inputs to the last fully connected layer in our experiments) $z(x)$ for each $x$, i.e., $\omega_{i,j} = \exp\left(-\|z(x_i) - z(x_j)\|^2 / 2\sigma^2\right)$, where $\sigma$ is the mean value of all the $k(k-1)/2$ pairwise distances.

In Figure 6, we show how the test set accuracy changes when increasing the number of training batches in each curriculum learning method on 4 datasets. In Figure 7, we report wall-clock training time on 2 datasets. *The results for other datasets can be found in the appendix*, together with the wall-clock time for (1) the entire training and (2) the submodular maximization part in DIHCL with diversity and MCL. The final test accuracy achieved by each method is reported in Table 1. DIHCL and its variants show signifi-

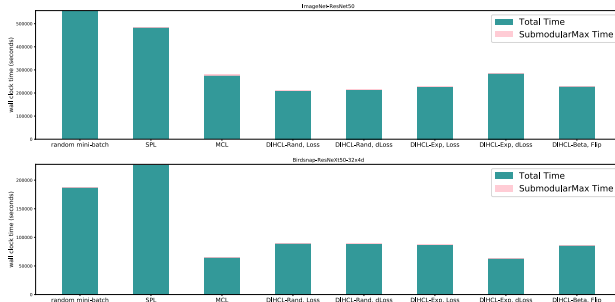

Figure 7: Wall clock training time of DIHCL (and its variants), SPL [25], MCL [52], and random mini-batch on ImageNet (top) and Food-101 (bottom).

cantly faster and smoother gains on test accuracy than baselines during training especially at earlier stages. DIHCL also achieves higher final accuracy and shows improvements in sample efficiency (meaning they reach their best performance sooner, after less computation has taken place). MCL can reach similar performance as DIHCL on some datasets but it shows less stability and requires more relative computation for submodular maximization. We also observe a similar instability of SPL. The reason is that, compared to the methods that use DIH, both MCL and SPL deploy instantaneous instance hardness (i.e., current loss) as the score to select samples, a measure that is more sensitive to randomness and perturbation that occurs during training. Compared to MCL and DIHCL, SPL and the random mini-batch curriculum method requires more epochs to reach their best accuracy, since they spend training effort on the memorable samples but lack repeated-learning of the forgettable ones. Although every variant of DIHCL achieves the best accuracy among all the evaluated methods on some datasets, DIHCL-Exp using loss and DIHCL-Beta using prediction flips, as the instantaneous hardness, exhibit advantages over the other DIHCL variants. Particularly, DIHCL-Exp with dLoss(metric (B)) is the best variant across datasets (achieving the top-2 performance on 8 out of the 11 datasets).

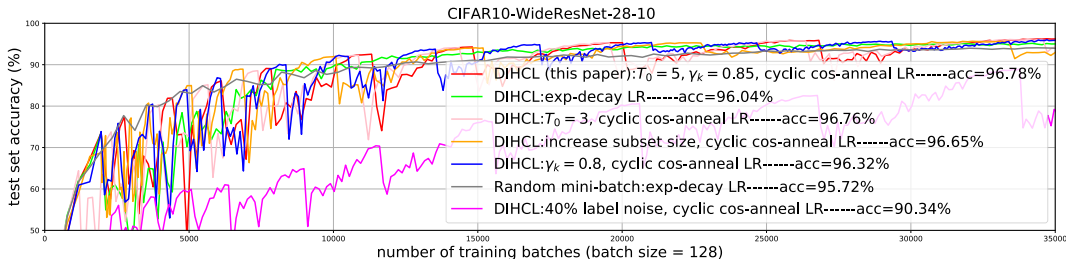

Figure 8: Comparison of DIHCL variants on training WideResNet-28-10 on CIFAR10.

We conduct an ablation study comparing several possible variants of DIHCL with their results reported in Figure 8. Specifically, we (1) change the cyclic cosine annealing learning rate to more commonly used exponentially decaying learning rate and compare DIHCL with random mini-batches; (2) reduce the number of warm starting epochs $T_0$ from 5 to 3; (3) increase the budget $k$ instead of decrease it in Line 12 of Algorithm 1 using $\gamma_k = 1.2$; (4) use a smaller discounting factor $\gamma_k = 0.8$; or (5) apply DIHCL on dataset containing $40\%$ noisy (wrong) labels. The results show that original DIHCL outperforms all the variants and is robust to noisy labels.

## 5  Conclusions

Inspired by human learning, we study a novel measure, "*dynamic instance hardness* (DIH)", which evaluates the hardness of a sample by using a running mean of an instantaneous hardness metric over training history. We find that DIH is a powerful tool to study the learning dynamics of DNNs and reveals several interesting properties of DNNs on individual samples during the course of training. Based on these properties, we develop *DIH guided curriculum learning* (**DIHCL**) in order to improve both the efficiency and final test-set performance without introducing notable extra costs, since DIH needs only to be lazily updated using training by-products. We demonstrate DIHCL's advantages over several recent CL methods and random baseline on 11 datasets.

## Broader Impact

We propose DIH guided curriculum learning as a general framework to improve efficiency for training machine learning models and their final performance. This potentially facilitates other applications and research that involve training machine learning models, e.g., using machine learning models to simulate high energy physics experiments, and automatically detect COVID-19 with CT scans. Moreover, as DIHCL is inspired by human learning, our results on machine learning models can also perhaps return the favor and be inspiring for those studying mechanisms behind true human learning. For example, we may use a metric similar to DIH to select human learning materials to test if better human learning efficiency can be achieved.

## Funding Disclosure

This research is based upon work supported by the National Science Foundation under Grant No. IIS-1162606, the National Institutes of Health under award R01GM103544, and by a Google, a Microsoft, and an Intel research award. It is also supported by the CONIX Research Center, one of six centers in JUMP, a Semiconductor Research Corporation (SRC) program sponsored by DARPA. Some GPUs used to produce the experimental results are donated by NVIDIA.

## Acknowledgments

We would like to thank NeurIPS area chairs and anonymous reviewers for their efforts in reviewing this paper and their constructive comments! We also thank Chandrashekhar Lavania, Lilly Kumari, and all the MELODI lab members for their helpful discussions and feedback.

## Footnotes

[2]We use $\eta_{t-1}$ instead of $\eta_t$ because $a_t(i)$ is computed based on $w_{t-1}$ before the weight update in step $t$.

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
