[Supplementary Material]

# A  Dynamic Instance Hardness (cont.)

In this section, we conduct two additional empirical studies about DIH on data with noisy labels and a smaller DNN as an extension of the one shown in the main paper.

Figure 9: **LEFT:** Averaged prediction-flip and **RIGHT:** losses (mean and std.) of the three groups of samples partitioned by a DIH metric (i.e.,running mean of prediction-flip) computed at epoch 10,40 and 60 during training WideResNet-28-10 on CIFAR10 with **random labels**. In this setting, the random (but wrong) labels will be remembered very well after some training, and DIH in early stages loses the capability to predict the future DIH, i.e., they can only reflect the history but not the future. This characteristic of DIH might be helpful to detect noisy data.

Figure 10: **LEFT:** Averaged prediction-flip and **RIGHT:** losses (mean and std.) of the three groups of samples partitioned by a DIH metric (i.e.,running mean of prediction-flip) computed at epoch 10, 40, 140 and 210 during training **a smaller CNN** on CIFAR10. It shows that the difference of memorable and forgettable samples is not sufficiently obvious until very late training epochs, e.g., after epoch-140.

First, **we conduct an empirical study of dynamic instance hardness during training a neural net on very noisy data**, as studied in [51] and [1]. In particular, we replace the ground truth labels of the training samples by random labels, and apply the same training setting used in Section 2. Then, we compute the running mean of prediction-flip for each sample at some epoch (i.e., 10, 40, 60), and partition the training samples into three groups, as we did to generate Figure 2. The result is shown in Figure 9. It shows 1) the group with the smallest prediction flip over history (left plot) is possible to have large but unchanging loss as shown in the right plot; and 2) the DIH in this case can only reflect the history but cannot predict the future. However, it also indicates that the capability of DIH to predict the future is potential to be an effective metric to distinguish noisy data or adversarial attack from real data. We will discuss it in our future work.

Table 2: Details regarding the datasets and training settings (#Feature denotes the number of features after cropping if applied), "lr_start" and "lr_target" denote the starting and target learning rate for the first episode of cosine annealing schedule, they are gradually decayed over the remaining epochs.

| Dataset | CIFAR10 | CIFAR100 | Food-101 | ImageNet | STL10 | SVHN |
|---|---|---|---|---|---|---|
| #Training | 50000 | 50000 | 75750 | 1281167 | 5000 | 73257 |
| #Test | 10000 | 10000 | 25250 | 50000 | 8000 | 26032 |
| #Feature | $(3, 32, 32)$ | $(3, 32, 32)$ | $(3, 224, 224)$ | $(3, 224, 224)$ | $(3, 96, 96)$ | $(3, 32, 32)$ |
| #Class | 10 | 100 | 101 | 1000 | 10 | 10 |
| #Epoch $T$ | 300 | 300 | 400 | 200 | 1200 | 300 |
| BatchSize | 128 | 128 | 80 | 256 | 128 | 128 |
| lr_start | $2 \times 10^{-1}$ | $2 \times 10^{-1}$ | $2 \times 10^{-1}$ | $2 \times 10^{-1}$ | $2 \times 10^{-1}$ | $2 \times 10^{-2}$ |
| lr_target | $5 \times 10^{-4}$ | $5 \times 10^{-4}$ | $1 \times 10^{-4}$ | $1 \times 10^{-4}$ | $5 \times 10^{-4}$ | $1 \times 10^{-3}$ |

Table 3: Details regarding the datasets and training settings (cont.)

| Dataset | Birdsnap | FGVCaircraft | StanfordCARs | KMNIST | FMNIST |
|---|---|---|---|---|---|
| #Training | 47386 | 6667 | 8144 | 50000 | 50000 |
| #Test | 2443 | 3333 | 8041 | 10000 | 10000 |
| #Feature | $(3, 224, 224)$ | $(3, 224, 224)$ | $(3, 224, 224)$ | $(1, 28, 28)$ | $(1, 28, 28)$ |
| #Class | 500 | 100 | 196 | 10 | 10 |
| #Epoch $T$ | 400 | 400 | 400 | 300 | 300 |
| BatchSize | 258 | 256 | 256 | 128 | 128 |
| lr_start | $4 \times 10^{-1}$ | $4 \times 10^{-1}$ | $4 \times 10^{-1}$ | $4 \times 10^{-2}$ | $4 \times 10^{-2}$ |
| lr_target | $1 \times 10^{-4}$ | $1 \times 10^{-4}$ | $1 \times 10^{-4}$ | $1 \times 10^{-3}$ | $1 \times 10^{-3}$ |

Second, **we change the WideResNet to a much smaller CNN architecture with three convolutional layers**[3]. We apply the same training setting used in Section 2. Then, we compute the running mean of prediction-flip for each sample at some epoch (i.e., 10, 40, 140, 210), and partition the training samples into three groups, as we did to generate Figure 2. The result is shown in Figure 10. Compared to DIH of training deeper and wider neural nets shown in Figure 2, the memorable and forgettable samples are indistinguishable until very late stages, e.g., Epoch-140. This indicates that using DIH in earlier stage to select forgettable samples into curriculum might not be reliable when training small neural nets. We will leave explanation of this phenomenon to our future works.

Moreover, we provide a comparison of the smoothness between DIH and instantaneous loss on individual samples in Figure 1. It shows that the DIH is a smooth and consistent measure of the learning/memorization progress on individual samples. In contrast, the frequently used instantaneous loss is much noisier, so selecting training samples according to it will lead to unstable behaviors during training. In Figure 11, we also provide a comparison of DIH and instantaneous loss on the two groups of samples in Figure 4, which shows a similar phenomenon.

## B  Experiments (cont.)

We use cosine annealing learning rate schedule for multiple epochs. The switching epoch between each two consecutive episode for different datasets are listed below.

- CIFAR10, CIFAR100, SVHN, KMNIST, FMNIST:
  $(5, 10, 15, 20, 30, 40, 60, 90, 140, 210, 300)$;
- STL10: $(20, 40, 60, 80, 120, 160, 240, 360, 560, 840, 1200)$
  $= 4 \times (5, 10, 15, 20, 30, 40, 60, 90, 140, 210, 300)$;
- ImageNet: $(5, 10, 15, 20, 30, 45, 75, 120, 200)$;
- Food-101, Birdsnap, FGVC-Aircraft, StanfordCars:
  $(10, 20, 30, 40, 60, 90, 150, 240, 400) = 2 \times (5, 10, 15, 20, 30, 45, 75, 120, 200)$;

We report how the test accuracy changes with the number of training batches for each method, and the wall-clock time for all the 11 datasets in Figure 15-18.

Figure 11: **Top:** DIH (running mean of loss) vs. **Bottom:** instantaneous loss of 10 samples randomly selected from the top 10k samples with the largest(red) and the smallest(blue) DIH at epoch 40 of training of WideResNet-28-10 on CIFAR10 (the same as Figure 4. It shows that for each individual sample from the two groups, DIH smoothly decreases while the corresponding instantaneous loss is much noisier.

Figure 12: Large Figure 8: Comparison of DIHCL variants for training WideResNet-28-10 on CIFAR10.

Figure 13: **TOP:** Averaged prediction-flip and **RIGHT:** losses (mean and std.) of the three groups of samples partitioned by a DIH metric (i.e., running mean of prediction flip) computed at epoch 40 when using **exponential decaying learning rate** (instead of cyclic cosine annealing rate) across epochs (cycles). DIH exhibits similar properties on identifying hard and easy samples for neural nets to learn.

Figure 14: Losses (mean and std.) of the three groups of samples partitioned by a DIH metric (i.e., running mean of prediction flip) computed at epoch 40 when 40% of labels are randomly changed to another wrong class (i.e., **40% symmetric noises on labels**. We also show the losses on the clean samples with correct labels and noisy samples with wrong labels, where the former exhibit lower DIH than the latter. Hence, DIH is robust to label noises and can identify the hard and easy samples, which are mainly composed of the clean and noisy data respectively in this scenario.

Figure 15: Training DNNs by using DIHCL (and its variants), SPL [25], MCL [52], and random mini-batch SGD on 3 datasets, i.e., CIFAR10, CIFAR100 and STL-10. We use "Diverse" to denote DIHCL that further reduces $S_t$ by applying submodular maximization for Eq. (3). We report how the test accuracy changes with the number of training batches for each method, and the (**log-scale**) wall-clock time for 1) the entire training and 2) the submodular maximization part in DIHCL with diversity and MCL.

Figure 16: Training DNNs by using DIHCL (and its variants), SPL [25], MCL [52], and random mini-batch SGD on 3 datasets, i.e., SVHN, Fashion MNIST and Kuzushiji MNIST. We use "Diverse" to denote DIHCL that further reduces $S_t$ by applying submodular maximization for Eq. (3). We report how the test accuracy changes with the number of training batches for each method, and the (**log-scale**) wall-clock time for 1) the entire training and 2) the submodular maximization part in DIHCL with diversity and MCL.

Figure 17: Training DNNs by using DIHCL (and its variants), SPL [25], MCL [52], and random mini-batch SGD on 3 datasets, i.e., ImageNet, Food-101 and Birdsnap. We report how the test accuracy changes with the number of training batches for each method, and the wall-clock time for 1) the entire training and 2) the submodular maximization part in MCL.

Figure 18: Training DNNs by using DIHCL (and its variants), SPL [25], MCL [52], and random mini-batch SGD on 2 datasets, i.e., FGVC Aircraft and Stanford Cars. We report how the test accuracy changes with the number of training batches for each method, and the wall-clock time for 1) the entire training and 2) the submodular maximization part in MCL.

Table 4: Test Acc (mean±variance) over 5 trials on two CIFAR datasets. It shows that the performance of DIHCL is stable and does not suffer from high variance.

| Curriculum | CIFAR10 | CIFAR100 |
|---|---|---|
| DIHCL-Rand, Loss | $96.74 \pm 0.04$ | $80.80 \pm 0.16$ |
| DIHCL-Rand, dLoss | $96.75 \pm 0.06$ | $80.73 \pm 0.21$ |
| DIHCL-Exp, Loss | $97.07 \pm 0.11$ | $82.31 \pm 0.24$ |
| DIHCL-Exp, dLoss | $96.44 \pm 0.10$ | $81.35 \pm 0.27$ |
| DIHCL-Beta, Flip | $96.48 \pm 0.04$ | $81.13 \pm 0.18$ |

## Footnotes

[3]The "v3" network from https://github.com/jseppanen/cifar_lasagne.