[Reviews · NeurIPS 2020]

Review 1

Summary and Contributions: 1. Introduces a new metric to estimate sample hardness based on the model historical predictions. 2. Presents some analysis showing the good properties of this new metric. 3. Proposes an algorithm to carry out curriculum learning with this new metric and empirically evaluates it on 11 datasets.

Strengths: 1. It is challenging to use curriculum learning to improve the performance on the full and well-labeled dataset. Many previous works failed to do so. This work does a pretty good job here by showing the consistent improvement across many data sets. 2. The proposed metrics and algorithm look reasonable. 3. The analysis on identifying memorable examples is interesting. It is even more interesting to see that the authors show the analysis both on clean and noisy training data.

Weaknesses: 1. Missing citations: It is similar to a prior work [ref1] which computes the variance of the past predictions; this idea was further developed into a general framework in [ref2]. These works should be discussed in the introduction. [ref4] should be mentioned in Section 3.2 because, to the best of my knowledge, it is the first work introducing random sampling in curriculum learning (called example dropout). 2. The statement may be revised that "updating instantaneous hardness typically requires extra inference steps of a model over all the samples" (Line 37). It may be true in the past but recent curriculum learning methods integrate this step inside mini-batch training and show a comparable convergence rate. See the SPADE algorithm in [ref2]. Besides, the authors did not talk about the extra GPU memory needed which appears to be the main reason that allows them to cut short in computation. 3. Is this metric specific to cyclic learning rate schedule, or it is general and can be used in other learning rate schedules, say piecewise exponential decay? If so, what was the performance compared to the cyclic learning rate. The reason to ask this question is because [ref3] shows that cyclic learning rates may lead to different learning patterns. [ref1] Chang, Haw-Shiuan, et al. "Active Bias: Training More Accurate Neural Networks by Emphasizing High Variance Samples." In NIPS 2017. [ref2] Jiang, Lu, et al. "Mentornet: Learning data-driven curriculum for very deep neural networks on corrupted labels." International Conference on Machine Learning. 2018. [ref3] Huang, Jinchi, et al. "O2U-Net: A Simple Noisy Label Detection Approach for Deep Neural Networks." Proceedings of the IEEE International Conference on Computer Vision. 2019. [ref4] Liang, Junwei, et al. "Learning to Detect Concepts from Webly-Labeled Video Data." IJCAI. 2016.

Correctness: Generally yes but with some missing references. The empirical method seems solid.

Clarity: Generally yes but 1. the experiment section is very dense and hard to parse. 2. the authors employ three metrics but keep referring them under the same name DIH.

Relation to Prior Work: Yes but with some missing references: [ref1] Chang, Haw-Shiuan, et al. "Active Bias: Training More Accurate Neural Networks by Emphasizing High Variance Samples." In NIPS 2017. [ref2] Jiang, Lu, et al. "Mentornet: Learning data-driven curriculum for very deep neural networks on corrupted labels." International Conference on Machine Learning. 2018. [ref4] Liang, Junwei, et al. "Learning to Detect Concepts from Webly-Labeled Video Data." IJCAI. 2016.

Reproducibility: Yes

Additional Feedback: The analysis of memorization on both clean and noisy data is very interesting. But for non-random noisy data, I wonder how DIH would perform when learning easy patterns first may not hold on real-world noisy data. This is a very new finding in [ref5]. [ref5]: Jiang, Lu, et al. "Beyond Synthetic Noise: Deep Learning on Controlled Noisy Labels." ICML 2020. ============= Post-rebuttal I have read the authors' responses and believe most of the points are sufficiently addressed. Given the authors will address the weaknesses in the revision (as promised in their rebuttal), I would like to see this paper get accepted.


Review 2

Summary and Contributions: I enjoy reading this paper. The paper proposed multiple metrics to understand the difficulty of each training sample, thereby achieving more efficient and robust training. Including the widely used instantaneous hardness (i.e., loss), the impact of two additional ways (I.e., loss change and prediction flip) is well studied. The authors point out that the simple use of the model's output results in a large difference between training sets selected over successive epochs; hence, exponential moving average is used for stability. In particular, the interpretations about their observations are interesting and have a potential impact for selecting batch samples adaptively.

Strengths: The authors' claim is well supported by extensive evaluation (or ablation in the main text). Also, they did their best to consider a lot of concerns, e.g., easy sample revisits. In addition, multiple variations were proposed and most of them achieved better performance compared with 3 baselines. Their evaluation on 11 datasets is also great.

Weaknesses: I have few concerns while reading this work. The baselines used for evaluation are limited; only two previous studies were compared. As far as I know, there is a lot of effort to achieve the same goal based on "sample-difficulty" or "sample-uncertainty" [1-4]. They proposed their own aspects to expedite the training process as well as improve the generalization capacity of the model. It is worthy to mention and compare. [1] Online Batch selection for faster training of neural networks (ICLRW'16) [2] Active Bias: Training more accurate neural networks by emphasizing high variance samples (NeurIPS'17) [3] Submodular batch selection for training deep mural networks (IJCAI'19) [4] Fixing mini-batch sequences with hierarchical robust partitioning (IJCAI'19) There is a conceptual difference in "increasing the subset size (previous studies)" vs "reducing the subset size (proposed one)". How about using the formal concept for your proposed method? In evaluation, It is unclear to judge the stability of your method without variance (or standard error) in Table 1. Data augmentation or other techniques for robustness was used (what kinds if used)? I am curious about the potential improvement when all those techniques are activated. The reported accuracy in Table 1 is not the state-of-the-art results; see the SOTA results in https://benchmarks.ai/

Correctness: All the claims were supported by extensive evaluation.

Clarity: This paper is well written and easy to follow. I enjoy reading this paper.

Relation to Prior Work: Only curriculum learning based methods were covered. There are lots of methods using other concepts.

Reproducibility: Yes

Additional Feedback: *** After reading the author feedback *** I read all the reviews and author feedback. I agree that the used exponential moving average is not that novel, but the authors did well to support their claims with extensive evaluation; the reported results seem to be quite strong. Also, a weakness I pointed out was narrow baseline methods (i.e., only CL methods were compared), but the authors additionally included other recent studies during the rebuttal, and the performance still significantly outperformed them. Hence, I'd like to see it at the conference.


Review 3

Summary and Contributions: This paper develops an adaptive curriculum based on dynamic instance hardness (DIH), which is the exponential moving average of a sample's instantaneous hardness. Empirical results show that the proposed DIH guided curriculum learning (DIHCL) outperforms existing CL methods on 11 datasets.

Strengths: This paper focus on developing an adaptive curriculum that leads to faster learning of more accurate models. To achieve this goal, this paper studies the dynamics of trainng a neural network, and proposed the dynamic instance hardness (DIH) as a metric for selecting hard samples in curriculum. DIH is the exponential moving average of a sample's instance hardness (e.g. loss, loss change, and prediction flip). This paper regarded samples with small DIH value as easy samples, and regarded samples with large DIH value as hard samples. By studying three properties of DIH, this paper introduces a natural curriculum learning strategy "DIH guided curriculum (DIHCL)" that keep training the model on hard samples and revisit easy samples less frequently. In each training step, DIHCL selects big-DIH samples for training and updates the DIH of the selected samples. The main stengths of this work are: 1. The proposed dynamic instance hardness (DIH) is a byproduct of back-propogation, therefore is comutational inexpensive.

Weaknesses: 1. Using the trick of exponential moving average is not novel. 2. The effect of the number of warm start epochs T0 is not studied. Any ablation study on T0? This also holds for other hyper-parameters such as the reducing factor \gamma_k of subset size. Minor: line 348: “it dynamic instance hardness (DIH)” --> "dynamic instance hardness (DIH)".

Correctness: Yes.

Clarity: Not really.

Relation to Prior Work: Yes.

Reproducibility: No

Additional Feedback: See weakness. Update: After reading the rebuttal and other reviews, I have raised my score since the rebuttal addressed most of my concerns. Thanks.


Review 4

Summary and Contributions: This paper designs a new curriculum learning strategy by using dynamic instance hardness to generate curriculums for learning tasks. The main idea is using the moving average of the instantaneous hardness as sample selection criterion during training for dynamically determining the samples involved into training process, and thus with good efficiency and stability. Experiments validate the effectiveness of the proposed method.

Strengths: Considering iteration dynamics in algorithm implementation makes the method more stable and consistent than previous CL methods. Greedily selecting curriculums in each iteration improves the efficiency of the algorithm. Experiments validate that the DIH can quickly converge to good values.

Weaknesses: How to tune some important hyper-parameters has not been clarified and analyzed. The experiments in supplementary material show that the method is somehow sensitive to noises, and thus is not sufficiently robust. Not provide theoretical evidences supporting the rationality of the proposed method.

Correctness: Yes.

Clarity: Yes.

Relation to Prior Work: Yes.

Reproducibility: Yes

Additional Feedback: In all, I think the DIH proposed in the paper is a rational curriculum specification strategy which should be helpful for learning tasks. My major concern is that the paper lacks a theoretical evidence to clarify the rationality of the proposed method, but contains many heuristic and intuitive explanations. It might be not easy to give the convergence or generalization theoretical results, but some theoretical understanding underlying the new method or intrinsic relationship to other known algorithms should be very helpful, just like those given in the original CL paper. This is the main reason that I could not give an evident acceptance score to this work. The following are several concerns I want the authors to further clarify in rebuttal. In the second experiment in supplementary material, it shows that the network architecture should have evident impact to the convergence of DIH. The carefully tuning for hyper-parameter T0 thus should be important. But the paper has not presented how to set this hyper-parameter in detail. In the first experiment in supplementary material, the results seem to show that the method is sensitive to noises, and thus is not sufficiently robust to real cases. In table 1, the results of STL10 and SVHN have not depicted with right colors. In the proof part in Appendix C, the contents in lines 578-580 and 602-604 are repetitive. %%%%%%%%%%%% After reading the rebuttal provided by the authors and all discussions by reviewers, I prefer to keep my original score. Thanks.

[Author Response · NeurIPS 2020]

We appreciate the reviewers' time and suggestions! We address them all and report new experimental results below.

*Reviewer 1*: • Missing citations:... We will cite the suggested citations and discuss their differences/relations with our method. Although DIH can be helpful to identify noisy data in noisy-label setting (ref.Middle plot in Fig. 1), our curriculum is not specifically designed for noisy labeled data and this paper mainly focuses on clean data setting. DIHCL still achieves 90.34% test-set accuracy under 40% symmetric label noise on CIFAR10 (ref.Top plot in Fig. 1).

• The statement may be revised that "updating instantaneous hardness typically requires extra inference steps of a model over all the samples" ... extra GPU memory... SPADE Alg in [ref2] samples a mini-batch and then selects samples within the mini-batch using the MentorNet. Comparing to DIHCL, SPADE incurs the following extra computational costs: (1) the feature extraction on the sampled mini-batch; (2) training of the MentorNet; (3) repeated training on well-learned and easy samples with correct labels. In addition, DIHCL can be incorporated with [ref2] as a predefined curriculum to train the MentorNet more efficiently. DIHCL requires extra GPU memory linear in the number of samples, which is negligible compared to the GPU memory for network training. We will add a discussion of the memory cost in the next version.

• Is the method specific to cyclic learning rate... DIHCL is applicable to other learning rate schedules.

Figure 1: Top: variants of DIHCL-Rand-dLoss; Middle & Bottom: dynamics of DIH-grouped samples under label noise and another lr schedule.

We report the result of DIHCL with a piecewise exponential decay learning rate in Fig. 1. DIHCL improves the test accuracy from 95.72% to 96.04% in this case. We also visualized the dynamics of samples partitioned by DIH at epoch-40 and it shows that the properties in Section 2 also hold for different learning rate schedules.

• Clarity We will simplify the experiment part and use better names for the DIH variants.

*Reviewer 2*: • Compare more baseline methods... Given the limited time for rebuttal, we compared DIHCL with [2]-[4] in Table 1 on CIFAR100 when used to train WideResNet-28-10. We will add a complete comparison in the next version.

• "Increasing the subset size (previous studies)" vs "reducing the subset size (as proposed)" It depends on one's preference for easy v.s. hard samples. If the curriculum always selects the easiest samples (e.g., SPL), the former should be used since only an increasing size can include harder but more informative samples in later stages. We reduce the subset size in DIHCL for two reasons: (1) in early stages, training on sufficient samples yields an accurate estimate of DIH since it is a time-moving average; (2) in later stages, we can reduce unnecessary training costs on easy/memorized samples, as suggested by observations in Section 2. In Fig. 1, we provide a comparison between the two and it shows that the former is slightly worse than the latter on the final test accuracy.

Table 1: [R2]Test accuracy (%) of WideResNet-28-10 on CIFAR100.

| | |
|---|---|
| DIHCL | 82.23 |
| [2] | 75.04 |
| [3] | 71.95 |
| [4] | 76.28 |

• Need variance for in Table 1. Given limited time, we report the mean±variance over 5 trials on two datasets in Table 2. We will add complete variance results in the next version.

• SOTA results with all techniques. We will try to compete with the baselines in the SOTA setting in the next version.

Table 2: [R3]Test Acc (mean±variance).

| Curriculum | CIFAR10 | CIFAR100 |
|---|---|---|
| DIHCL-Rand, Loss | $96.74 \pm 0.04$ | $80.80 \pm 0.16$ |
| DIHCL-Rand, dLoss | $96.75 \pm 0.06$ | $80.73 \pm 0.21$ |
| DIHCL-Exp, Loss | $97.07 \pm 0.11$ | $82.31 \pm 0.24$ |
| DIHCL-Exp, dLoss | $96.44 \pm 0.10$ | $81.35 \pm 0.27$ |
| DIHCL-Beta, Flip | $96.48 \pm 0.04$ | $81.13 \pm 0.18$ |

*Reviewer 3*: • Effects of $T_0$ and $\gamma_k$... $T_0$ is necessary to get an early estimate of DIH and is supported by the theoretical analysis in the Appendix. But a small $T_0 = 3$ suffices to get a stable training because: after the $T_0$ warm-starting epochs, we start from all samples (exploration) and gradually reduce the subset size. In Fig. 1, we compare $T_0 = 5$ and $T_0 = 3$: they produce similar test accuracy. We also compare $\gamma_k = 0.85$ and $\gamma_k = 0.8$: reducing the subset size too fast ($\gamma_k$ is too small) will degrade the accuracy.

*Reviewer 4*: • $T_0$ and hyper-parameter tuning. Please also see our reply to Reviewer 3. DIHCL is not sensitive to the choice of $T_0$ since the earlier epochs after warm starting select almost all samples (exploration), which keeps the DIH's estimation accurate. We do not have computation for a full grid search of all hyperparameters so there could exist better choices. Line 307-312 details how we selected the hyperparameters.

• Method sensitive to noise. The noisy-label experiments in the Appendix use a 100% noise setting, i.e., all the labels are randomized and wrong. The purpose is to show that the pattern of DIH is very different on clean and noisy data, so we can use DIH as an indicator of label noise. In Fig. 1, we report the results under 40% symmetric label noise. It shows (1) DIH contains critical information to identify the noisy-labeled data, and (2) DIHCL is robust to label noise and achieves 90.34% test-set accuracy under 40% symmetric label noise on CIFAR10.

[Meta-Review · NeurIPS 2020]

Thank you for submitting your work to NeurIPS. All reviewers were enthusiastic about the paper, and I am happy to accept it. Strong empirical results and practicality of the method were appreciated by the reviewers. Expert reviewers found the method sufficiently novel for publication. However, the fact that other papers use the exponential moving average (e.g. http://proceedings.mlr.press/v80/jiang18c/jiang18c.pdf) has to be discussed much more clearly. Please make sure that you include in the introduction (not just related work) section a detailed discussion on prior work that was flagged by R1. Please also make sure to include a full comparison with prior work (as in Table 1 in the rebuttal).